# A Study on Wheel Member Condition Recognition Using Machine Learning (Support Vector Machine)

**DOI:** 10.3390/s23208455

**Published:** 2023-10-13

**Authors:** Jin-Han Lee, Jun-Hee Lee, Kwang-Su Yun, Han Byeol Bae, Sun Young Kim, Jae-Hoon Jeong, Jin-Pyung Kim

**Affiliations:** 1Busan Transportation Corporation, Busan 47353, Republic of Korea; jin2023145@humetro.busan.kr (J.-H.L.); longsooa@humetro.busan.kr (K.-S.Y.); 2School of Software Engineering, Kunsan National University, Gunsan 54150, Republic of Korea; zmdzmd0541@naver.com; 3School of Mechanical Engineering, Kunsan National University, Gunsan 54150, Republic of Korea; 4Global Bridge Co., Ltd., Incheon 21990, Republic of Korea

**Keywords:** recognizing condition algorithm, machine learning algorithm, wheel, tire

## Abstract

The wheels of railway vehicles are of paramount importance in relation to railroad operations and safety. Currently, the management of railway vehicle wheels is restricted to post-event inspections of the wheels whenever physical phenomena, such as abnormal vibrations and noise, occur during the operation of railway vehicles. To address this issue, this paper proposes a method for predicting abnormalities in railway wheels in advance and enhancing the learning and prediction performance of machine learning algorithms. Data were collected during the operation of Line 4 of the Busan Metro in South Korea by directly attaching sensors to the railway vehicles. Through the analysis of key factors in the collected data, factors that can be used for tire condition classification were derived. Additionally, through data distribution analysis and correlation analysis, factors for classifying tire conditions were identified. As a result, it was determined that the *z*-axis of acceleration has a significant impact, and machine learning techniques such as SVM (Linear Kernel, RBF Kernel) and Random Forest were utilized based on acceleration data to classify tire conditions into in-service and defective states. The SVM (Linear Kernel) yielded the highest recognition rate at 98.70%.

## 1. Introduction

The wheels of a railway vehicle are the most important part in relation to safety in railroad operation. They are structured to have a specific shape that prevents derailment, which can be caused by wearing of the wheel due to continuous friction with the rail, and at the same time to ensure stable riding comfort. However, in general, the wheels of railway vehicles receive a continuous contact load with the rail during the operation of the railway vehicle and, accordingly, damage to the wheel tread due to the continuous and ongoing contact load generates an impact load during operation. Therefore, it affects the truck parts where the wheels are installed, causing wheel damage and derailment, which can pose a risk to human safety, as well as a negative impact on customer convenience and reduced transportation due to delays and reduced ride comfort [1,2,3,4].

Currently, the wheel management of railway vehicles is limited to a follow-up inspection of the wheels whenever physical phenomena, such as abnormal vibration and noise caused during the operation of railway vehicles, are observed. Accordingly, there is a constant risk of human casualty and damage to railway vehicles due to wheel burnout. Furthermore, as maintenance is performed after the follow-up inspection of the wheel, the maintenance cost also increases, which is accompanied by economic loss.

The maintenance system of Korean railway vehicles has reached a level where it can utilize RCM (Reliability Centered Maintenance) technology based on statistics through the establishment of an information system. However, since it is difficult to respond to accidental failures, continuous development for preventing failures during operation is necessary for core devices, such as motor blocks, bearings of axles, and wheels, which directly causes passenger safety and operation delays [5,6].

In general, ultrasonic technology, infrared cameras, and magnetic field methods are used to determine wheel abnormalities. There is an acoustic vibration technology that analyzes wheel vibrations and acoustic vibrations of railway vehicles to detect abnormalities and vibration changes due to damage or imbalance of wheels. There is also a technology that uses image processing technology to photograph and analyze the wheel surface of a railway vehicle to determine whether there is an abnormality [7,8,9,10].

However, most ongoing wheel state recognition research is based on steel wheels. In addition, there is a lack of research on machine learning-based tire condition recognition and tire condition recognition systems. Research on key element analysis of tire sensor data and research on rubber wheels must be carried out in order to apply to machine learning models.

Figure 1 represents a flowchart illustrating the method for wheel condition recognition.

This paper proposes a system for recognizing and predicting the condition of railroad wheels using machine learning technology as one of the methods to address the aforementioned problem. In this paper, a machine learning algorithm trained on optimized training data is employed to monitor the condition of railroad vehicle wheels and predict anomalies in advance. This system goes beyond traditional post-inspection methods by analyzing data in real-time to detect and predict abnormalities.

This paper provides a device and method to enhance the learning and prediction performance of the machine learning algorithm by selectively identifying and specifying training data through correlation analysis. It determines influencing factors that cause anomalies in railroad wheel axles moving along the rail. Using the identified final influencing factors, a machine learning algorithm is trained, and real-world data related to wheel axles corresponding to these final influencing factors are applied to predict anomalies. To collect suitable data, sensors were attached to trains, marking the first system of its kind in the relevant field, and data were collected from trains operating on the Busan Line 4 subway in South Korea. Based on the collected data, machine learning was performed by creating training datasets. Therefore, in this paper, in order to accurately classify the condition of tire wear, sensing data targeting Busan light rail wheels (rubber tires) were used. The data consisted of factors including 3-axis acceleration, temperature, and pressure, which have never been applied to tire condition classification before. We propose a method to classify the wear state of tires of light rail trains based on machine learning by measuring them.

This paper consists of 6 chapters as follows: Section 2 provides information on how to predict wheel abnormalities in existing railway vehicles; Section 3 explains factor analysis related to tire recognizing condition; Section 4 explains how to build tire recognizing condition learning data; Section 5 describes the tire recognizing condition algorithm; and finally, Section 6 offers conclusions.

## 2. Technological Trends in Predicting Abnormalities in Railway Vehicle Wheels

In this chapter, we look at studies on the status recognition of subway wheels and related studies on the status recognition of rubber tires.

In “Infrared diagnostics of cracks in railway carriage wheels”, a method for detecting cracks in railway wheels was developed using infrared cameras and thermal imaging cameras. It was based on recording the change in surface temperature distribution of the wheel disc while the disc of the railway wheel was heated [11].

“Supplementary magnetic tests for railway wheel sets” proposed a further step in the diagnosis of railway wheelsets based on the magnetic mechanic effect and magnetic processes that occur in the material because of aging due to the magnetic properties of steel [12].

“A new measuring method of wheel–rail contact forces and related considerations” proposed a new method for measuring wheel–rail contact forces, using non-contact gap sensors without special wheelsets equipped with strain gauges, slip rings, or telemeters. It was measured from lateral distortion [13].

In “Acoustic emission monitoring of wheel sets on moving trains,” an acoustic emission monitoring system for wheels of moving trains and trams was proposed using AE (Acoustic emission) sensors mounted on rails [14].

“Profile parameters of wheelset detection for high speed freight train” presented a system to obtain accurate wheelset profile parameters by using lasers and high-speed cameras and applying non-contact light section technology [15].

In “Wheel defect detection with machine learning”, we proposed two machine learning methods to automatically detect wheel defects based on the wheel vertical force measured by the sensor. SVM is used to classify time series data, and an artificial neural network is provided for image classification [16].

“Machine Learning-Based Wheel Monitoring for Sapphire Wafers” used machine learning classification algorithms (k-NN, ANN, SVM) to extract signals and characteristic parameters of the wheel during the polishing process using AE signals and radial/axial vibration signals [17].

“A study on the fatigue characteristics and life prediction of the tire sidewall rubber” proposed a life prediction equation using strain energy density obtained by performing tensile and fatigue tests on tire sidewall compounds. The applicability of the research results to low fuel efficiency tires was examined by converting the fatigue life of the sidewall rubber into the expected mileage of the tire [18].

“Definition of a new predictor for multiaxial fatigue crack nucleation in rubber” derived new predictive variables for fatigue crack nucleation of rubber. This is triggered by fatigue and motivated by microscopic mechanisms developed in the framework of Configurational Mechanics [19].

Tire wear recognition technology is being actively researched as one of the core technologies of intelligent tire systems. Most studies use a method of predicting durability by analyzing rubber properties through simulation using tires attached to actual cars and testing the fatigue level of its elasticity. However, accurate predictions are difficult depending on the type of vehicle, driving habits, road conditions, etc., and it is difficult to deliver status information to drivers in real-time, such as remaining tire life [18,19].

In this study, sensing data composed of factors including 3-axis acceleration, temperature, and pressure were measured for Busan light rail wheels (rubber tires) that adopted an unmanned driving system to accurately classify the state of tire wear. We propose a method to classify the wear condition of tires of light rail trains based on machine learning, which has never been applied to tire condition classification, and achieved high accuracy [20].

## 3. Analysis of Major Factors Based on Machine Learning

### 3.1. Analysis of Factors Related to Tire Recognizing Conditions for Light Rail Transit

To analyze the factors that affect tire wear, a method for calculating the degree of influence by considering the characteristics of related factors and data analysis for related studies is used. The factors are classified into driving/non-driving conditions and described in Table 1. A factor is considered if it can be used for recognizing tire condition according to light rail transit tire application and data collection [21].

Table 2 is a description of the experimental environment for collecting acceleration data, and KUMHO TIRE helped with the experiment. Using laboratory measurement data provided by KUMHO TIRE, factors that can affect tire condition are identified by the characteristics (pattern) of tires currently in use and tires in an old state. Acceleration factor analysis is used and tire driving data collection conditions is reviewed. The learning training data to be provided to the state classification model is refined.

The tire data for factor analysis is for 180,005 tires currently in use and 180,003 tires in the old state (worn). An analysis is conducted on tire state classification using acceleration factor patterns according to changes in light rail transit driving speed (test speed) and light rail transit load (test load).

### 3.2. Tire Currently in Use/Tires in the Old State (Worn) Characteristic Analysis of Tire Acceleration Factor by Condition

By analyzing the distribution of acceleration (X, Y, Z) data inside the tire according to the change in load and speed of the tire currently in use and tires in an old state, a factor highly correlated with the tire condition is sought. 

Figure 2 Shows the acceleration distribution according to load and speed changes of current rubber tires. In the case of tires currently in use, there is little change in acceleration (X, Y, Z) data distribution according to load change. Therefore, it is judged that there is a limit to the state classification of the tire using the relationship between the load status of the light rail transit and the acceleration inside the tire. Throughout the acceleration (X, Y, Z) data distribution according to the speed change, the size and density of the accelerations increase as the speed increases. In particular, the change (size, distribution) of the *Z*-axis occurs the most.

Figure 3 Shows the acceleration distribution according to load and speed changes of high-quality old tires. In the case of tires in the old state, it is confirmed that the change in acceleration (X, Y, Z) data distribution according to the load change is not large. In all the tires currently in use/tires in the old state, the influence of the acceleration factor according to the load change in the driving state is judged to be low.

In the driving environment, when the load is the same and the speed is changed, the data density according to the size of the *Z*-axis tends to decrease more significantly in tires in their old state than in tires currently in use. Through this, it is judged that the degree to which the *Z*-axis is affected by the tire’s state in the acceleration factor is large.

The tire acceleration measurement data are affected by the tire’s state. Using data distribution characteristic analysis of the acceleration factor, it is possible to check if it is applicable to the recognizing condition algorithm.

It is necessary to analyze the acceleration factor of the data acquired in the actual light rail transit driving environment and confirm if it is a factor that can classify the tire condition in the same way as the acceleration measurement data analysis result in the test environment.

### 3.3. Tires Currently in Use/Tires in the Old State (Worn) Correlation Analysis of Tire Acceleration Factor for Each State

Figure 4 and Figure 5 analyze the relationships between accelerations (X, Y, Z) in response to variations in load and speed, as well as the relationships among accelerations (X, Y, Z) through correlation analysis.

The accelerations (X, Y) of the tire currently in use increase in magnitude in proportion to each other with a positive correlation and are not significantly affected by load changes. Accelerations (X, Z) and accelerations (Y, Z) show a negative relationship, are inversely proportional, and are less affected by load changes. Since there is almost no change in the acceleration (X, Y, Z) correlation coefficient according to the load change, it is judged that the influence of the load change on each axis of acceleration is small.

In the case of the same load, as the test speed increases, the correlation coefficient values of the X and Y axes decrease, so the degree of influence decreases as the driving speed increases. However, the *Z*-axis shows a high correlation coefficient regardless of the change in test speed, so among the acceleration factors it is a factor that is greatly affected by driving.

Through the correlation analysis of the acceleration factor of tires in the old state, the correlation coefficient changes of each acceleration axis show a change similar to that of the tire currently in use, and it is equally confirmed that the effect is greater in relation to speed than load. Even when subjected to the same load, the *Z*-axis shows a high correlation coefficient regardless of the test speed change, as in the tire currently in use, so it can be seen that the degree of influence related to driving is a large factor. Therefore, through the coefficient value obtained by correlation analysis of the X, Y, and Z axes of the weight, the factor with the greatest influence in relation to driving is the *Z*-axis. When classified into tires currently in use and tires in the old state, the correlation coefficient value remains high in tires in the old state, so it seems to be applicable to state classification.

### 3.4. Light Rail Transit Driving Measurement Data Factor Analysis

Table 3 is a test environment to analyze what can be applied as a factor of the tire recognizing condition algorithm through the analysis of light rail transit driving measurement data provided by Busan Transportation Corporation in Busan, South Korea.

Table 4 selected candidate factors that are expected to be highly related to tires, excluding information for checking the state of sensors among the factors of light rail transit driving measurement data (153 channels). In addition, through data analysis, it was confirmed whether a factor can be applied as a major factor in tire recognizing condition.

The analysis process is performed to determine whether to apply the selected candidate factors as a recognizing condition factor by checking the degree of influence through correlation analysis with the tire state according to the tread depth.

Figure 6 shows the analysis of the correlation between tire measurement factors. Correlation analysis is conducted for each measured position (wheel) of the factors to be applied to the recognizing condition algorithm for recognizing the state (safety, warning, and danger) based on the tire tread. Based on the high correlation coefficients displayed by TTPMS_BAT_V, TTPMS_P, TTPMS_P_GAUGE, and TTPMS_T1 to 16, it is determined that they can be applied as significant factors.

Figure 7, Figure 8 and Figure 9 perform the confirmation process for factors that can classify tire conditions through measurement data distribution of major factors (axle acceleration and tire internal temperature/pressure).

The main factors of the measured data show a different form when checking the pattern classified according to the tire state and can be used as a recognizing condition factor using these characteristics.

## 4. Creating Learning Data for Tire Recognizing Condition Based on Machine Learning

### 4.1. Light Rail Transit Measurement Data Conversion and Learning Data Annotation

From the raw data measured through light rail transit driving, data acquired at similar times are selected in consideration of the external environment (season, time, etc.) when driving, and converted into a file (CSV format) to be linked with the program.

As shown in Table 5, for the route driven by light rail transit, the data were selected from two round trips (Anpyeong → Minam and Minam → Anpyeong) for each tire condition, and learning data were created by merging the four sections.

The raw data including factors measured through light rail transit driving were divided into tire conditions (safety, warning, and danger) according to the tread depth, and processed as data having the corresponding state as the target value light rail transit. Table 6 shows the learning data of the tire state recognition algorithm obtained from the light rail.

### 4.2. Light Rail Transit Instrumentation Data Sampling and Outlier Removal

The tires of the Busan Metro Line 4 electric train are equipped with three-axis (X, Y, Z) acceleration sensors inside the tires. Additionally, temperature and pressure sensors are attached to the four tires of the locomotive and trailer car. Four three-axis acceleration sensors are also installed near the tires of the locomotive and trailer cars.

Furthermore, reflective plate-type (non-contact laser) sensors are attached to the four points on the wheels of the locomotive and trailer cars, allowing conversion to angular velocity, speed, and distance measurements. To ensure temporal synchronization between vehicle information (speed, distance, and position) and collected data (pressure, temperature, acceleration, etc.), an IMU sensor is installed at one point in the vehicle’s underside. This setup is integrated with a program capable of data analysis and synchronization.

Figure 10 shows the location of the measurement sensors installed on the light rail.

Furthermore, to synchronize the sensor data collected during the operation of the metro, the sampling rate of the acceleration sensors is set to 1000 Hz, while the sampling rate for the internal tire temperature/pressure sensors is set to 100 Hz due to their different requirements. The minimum synchronization frequency of 100 Hz is achieved.

Table 7 is comparing the learning data configured through sampling and the axle acceleration graph pattern of the original data, the loss was confirmed, and the usability was determined. There was no significant difference, so it was used.

Figure 11 is a comparison between the original learning data and the learning data obtained through sampling.

Among the factor data measured through light rail transit driving, the data measured with an operating speed of 60 km/h or more and the data with a tire internal temperature exceeding ±100 °C were removed and shown in Table 8 and Table 9.

## 5. Development of a Machine Learning-Based Tire Recognizing Condition Algorithm

### 5.1. Machine Learning Algorithm

Figure 12 is a block diagram of the state recognition device for the rubber wheels of the light rail proposed in this paper.

The quality of the learning data is increased by selectively collecting data expected to cause wheel abnormalities and the learning data (i.e., wheel abnormalities) that will be used to train the machine learning algorithm through quantitative data analysis techniques. By selecting and specifying the final influencing factors that cause wheel abnormalities, the load required for learning the machine learning algorithm was reduced and prediction performance was improved.

Table 4 shows the collected status data. It is possible to optimize the number of final influencing factors to prevent overfitting of the machine learning algorithm that may occur when using all collected state data as machine learning training data, and to use a quantitative method to optimize the number of final influencing factors. As a result of analyzing the correlation in Figure 6, the tire’s internal temperature, internal pressure, and three-axis acceleration are set as main state data as it is the data with a Pearson correlation coefficient greater than the predefined standard value for the tire tread depth. Through a time series correlation analysis algorithm such as the EDA (Exporatory Data Analysis) algorithm, it is determined whether the time series change pattern between each datum is consistent. As shown in Figure 7, Figure 8 and Figure 9, it can be seen that the time series change patterns are consistent. Through this process, the final factors are determined and a machine learning algorithm is learned and operated to predict wheel abnormalities. In this paper, Random Forest and SVM (Linear Kernel, RBF Kernal) are used.

### 5.2. Machine Learning-Based Tire Recognizing Condition Algorithm Using Acceleration Measurement Data

To determine machine learning-based tire tread wear, feature data are created by converting vibration components of the acceleration sensor inside the tire into the frequency domain.

SVM is used for classification and regression analysis and is a type of supervised learning algorithm for pattern recognition and data analysis. SVM is based on the concept of finding a hyperplane that best separates data points into two or more classes. When data cannot be separated linearly, a technique called the kernel trick is used. The kernel trick maps data points to a high-dimensional space that can be separated by a hyperplane. Kernels include linear, polynomial, and RBF (Radial Basis Function) kernels [22].

For SVM-based tire state recognition using acceleration data, an SVM classification model tested the classification accuracy using tires currently in use and tires in the old state. In addition, the tire acceleration data targeted 180,005 tires currently in use and 180,003 tires in the old state (worn), setting tires in the old state to 0 and tires currently in use to 1, depending on the state of the tire. Figure 13 shows tire condition classification based on SVM using acceleration data.

Random Forest, a machine learning algorithm, can be used in the field of data classification and regression by generating a final learner by linearly combining it after generating several decision tree learners that learn arbitrarily. To prevent overfitting, it is an algorithm of the principle of randomly selecting an optimal reference variable. Random Forest is used because it is effective in processing large amounts of data and has the advantage of high model accuracy by minimizing overfitting problems [17]. Figure 14 shows tire condition classification based on RF using acceleration data.

### 5.3. Machine Learning-Based Tire Recognizing Condition Algorithm Using Light Rail Transit Driving Measurement Data

Based on the SVM model, a tire condition (safety, warning, and danger) classification experiment is conducted, and light rail transit driving data measurement is conducted on 1,567,706 train data and 732,469 test data (safety: 15 mm, warning: 8 mm, and danger: 1.6 mm). As a result of predicting tire state (safety, warning, and danger) based on the SVM model, the accuracy of the Linear Kernel and RBF Kernel are measured at 98.70% and 96.55%, respectively. Table 10 and Table 11 tabulate the results obtained using the SVM algorithm of Linear Kernel and RBF Kernel, respectively. 

Based on the Random Forest model, the tire state (safety, warning, and danger) classification experiment is conducted, and the light rail transit driving data measurement is conducted for 1,566,871 train data and 732,469 test data (safety: 15 mm, warning: 8 mm, and danger: 1.6 mm). As a result of predicting tire state (safety, warning, and danger) based on the Random Forest model, the accuracy was measured at 89.68%. Table 12 tabulates the results obtained using the Random Forest algorithm.

Looking at the tire state recognition results based on SVM (Linear Kernel) and SVM (RBF Kernel), it can be seen that the recognition rate of the tire state safety and warning is high, but the recognition rate of the tire state danger is relatively low.

Looking at the Random Forest-based tire state recognition results, it can be seen that the warning tire state show a high recognition rate, but the tire state safety and danger recognition rates are relatively low.

Table 13 compares the performance of the machine learning-based tire recognizing condition algorithm using light rail transit driving measurement data. The tire state estimation results are compared with the evaluation indexes of Accuracy (accuracy), Recall (recall rate), Precision (precision), and F1 Score.

## 6. Conclusions

This paper aims to improve operational efficiency, reduce maintenance costs, and ensure the safe operation of electric trains by implementing proactive maintenance measures through tire condition recognition (safe, warning, and danger) during train operation. To achieve this goal, actual equipment, including tire pressure/temperature sensors, internal acceleration sensors, vehicle axis vibration sensors, vehicle speed, and vehicle motion sensors, was installed on the tires of Busan Subway Line 4, and data were collected during the train’s operation. There were difficulties in installing measurement sensors inside a moving tire. The rubber tires of light rail wheels have iron auxiliary wheels inside the tires, so if the tires are damaged during operation, the vehicle can be restored through the auxiliary wheels. There was difficulty installing the sensor because the gap between the auxiliary wheels and tires was narrow. The collected data was correlation analyzed to identify key factors that could be used for tire condition recognition. As a result, it was determined that the *Z*-axis acceleration data had a significant influence on tire condition, depending on its state. Data distribution analysis and correlation analysis were conducted on the accelerations (X, Y, Z) to analyze the relationship between acceleration components and the relationship between different acceleration axes for currently in use or old state (worn) tires. To determine the applicability of the tire condition recognition factors, correlation analysis of measurement factors for four tires of a single electric train was performed using data collected during the operation of the Busan Metro Line 4. Acceleration of the axle and tire internal temperature/pressure data were examined as potential factors for classifying tire conditions based on data distribution. For the generation of learning data, two round trips of data from four sections of the Line 4 were merged, resulting in a total of 1,567,706 training data samples and 732,469 test data samples. Preprocessing was applied to the data, including sampling synchronization and outlier removal. To classify tire conditions, machine learning algorithms such as Support Vector Machine (SVM) and Random Forest (RF) were utilized, using acceleration, temperature, and pressure data. Initially, classification models were experimented with for the currently in use and old state (worn) tires, and the accuracy was measured. Among the three models, the Random Forest achieved a classification accuracy of 89.68%. For SVM, the Linear Kernel and RBF Kernel achieved classification accuracies of 98.70% and 96.55%, respectively, with the Linear Kernel showing the highest accuracy of 98.70%. This study enhanced the learning and prediction performance of machine learning algorithms using temperature, pressure, and three-axis acceleration data, and the frequency-domain transformed features of acceleration sensor vibrations. It not only provided a method for predicting anomalies in railway vehicle wheels in advance, rather than relying on post-checks after anomalies occur, but also demonstrated the potential for future applications and scalability in the expanding field of metro railways. There were limitations to predictions regarding tire cracks and damage. Data can only be collected when cracked or damaged tires are operated by installing cracked or damaged tires on a train. However, there is a limit to installing cracked or damaged tires on a light rail train that must handle passengers and operate safely. In future research, we plan to conduct additional research using light rail that does not handle passengers, increase the accuracy of state recognition through wavelet noise reduction, and conduct additional research on tire state recognition using deep learning. An updated paper will be submitted as part of our ongoing efforts to predict tire condition.

## Figures and Tables

**Figure 1 sensors-23-08455-f001:**
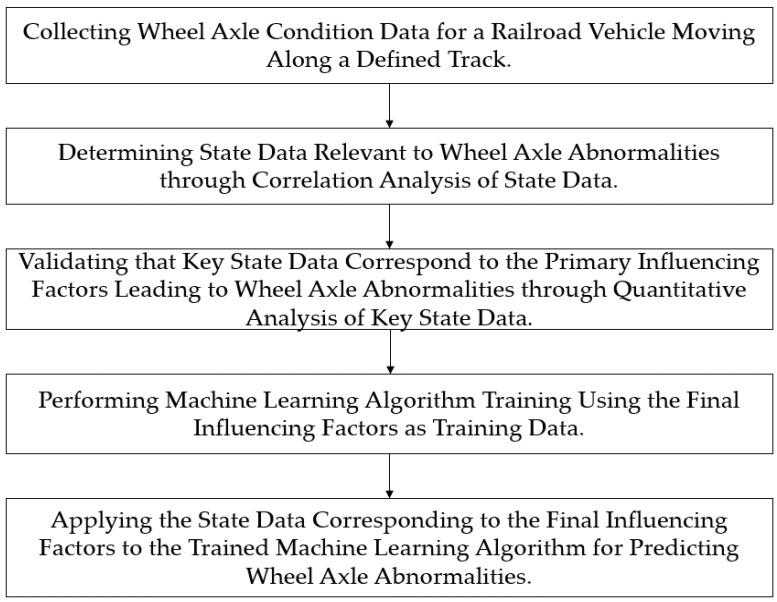
Flowchart for Explaining the Method of Wheel Condition Recognition.

**Figure 2 sensors-23-08455-f002:**
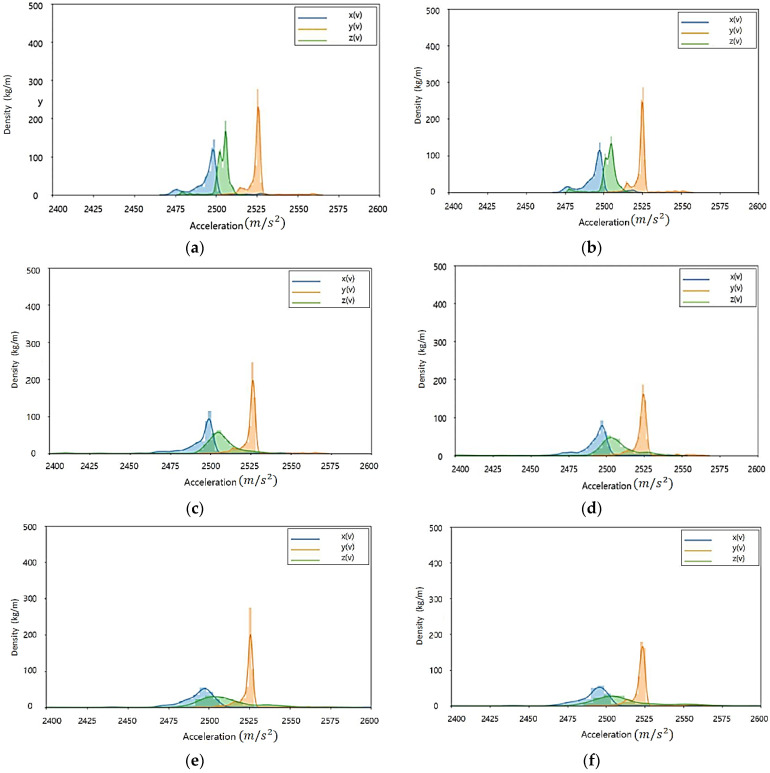
Acceleration distribution according to load and speed change of tire currently in use: (**a**) Test load: 3000 kgf, test speed: 20 km/h; (**b**) Test load: 4500 kgf, test speed: 20 km/h; (**c**) Test load: 3000 kgf, test speed: 40 km/h; (**d**) Test load: 4500 kgf, test speed: 40 km/h; (**e**) Test load: 3000 kgf, test speed: 60 km/h; (**f**) Test load: 4500 kgf, test speed: 60 km/h.

**Figure 3 sensors-23-08455-f003:**
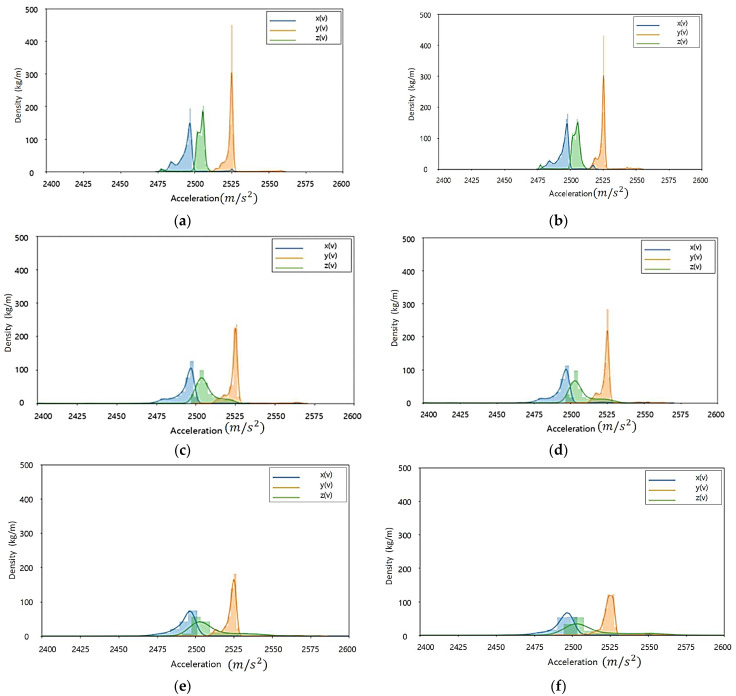
Acceleration distribution according to load and speed change of tires in the old state: (**a**) Test load: 3000 kgf, test speed: 20 km/h; (**b**) Test load: 4500 kgf, test speed: 20 km/h; (**c**) Test load: 3000 kgf, test speed: 40 km/h; (**d**) Test load: 4500 kgf, test speed: 40 km/h; (**e**) Test load: 3000 kgf, test speed: 60 km/h; (**f**) Test load: 4500 kgf, test speed: 60 km/h.

**Figure 4 sensors-23-08455-f004:**
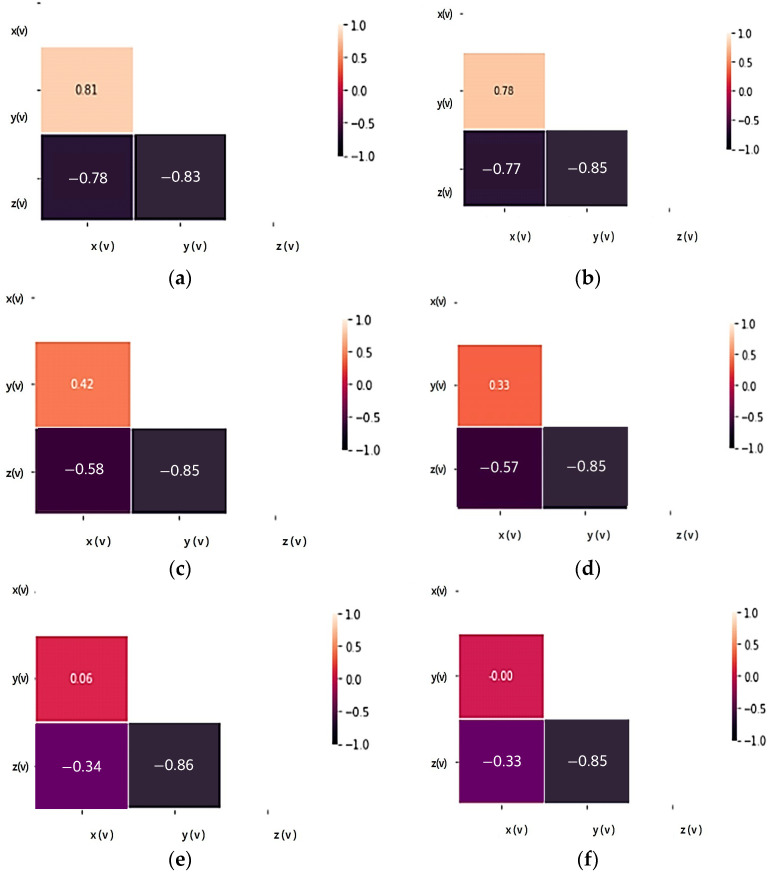
Acceleration correlation analysis according to load and speed change of tire currently in use: (**a**) Test load: 3000 kgf, test speed: 20 km/h; (**b**) Test load: 4500 kgf, test speed: 20 km/h; (**c**) Test load: 3000 kgf, test speed: 40 km/h; (**d**) Test load: 4500 kgf, test speed: 40 km/h; (**e**) Test load: 3000 kgf, test speed: 60 km/h; (**f**) Test load: 4500 kgf, test speed: 60 km/h.

**Figure 5 sensors-23-08455-f005:**
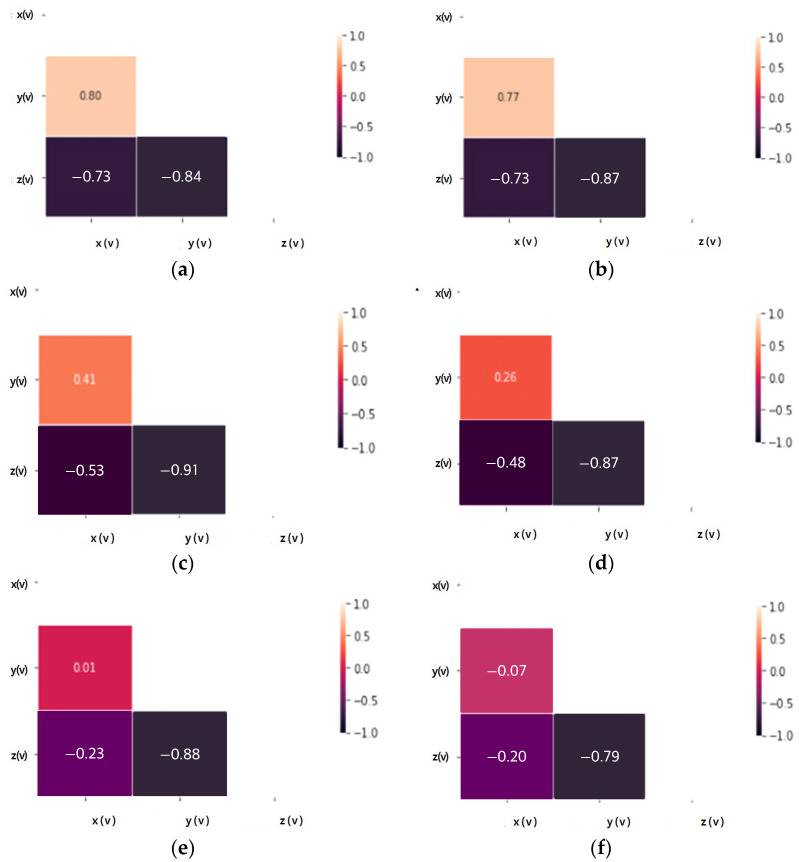
Acceleration correlation analysis according to load and speed change of tires in the old state: (**a**) Test load: 3000 kgf, test speed: 20 km/h; (**b**) Test load: 4500 kgf, test speed: 20 km/h; (**c**) Test load: 3000 kgf, test speed: 40 km/h; (**d**) Test load: 4500 kgf, test speed: 40 km/h; (**e**) Test load: 3000 kgf, test speed: 60 km/h; (**f**) Test load: 4500 kgf, test speed: 60 km/h.

**Figure 6 sensors-23-08455-f006:**
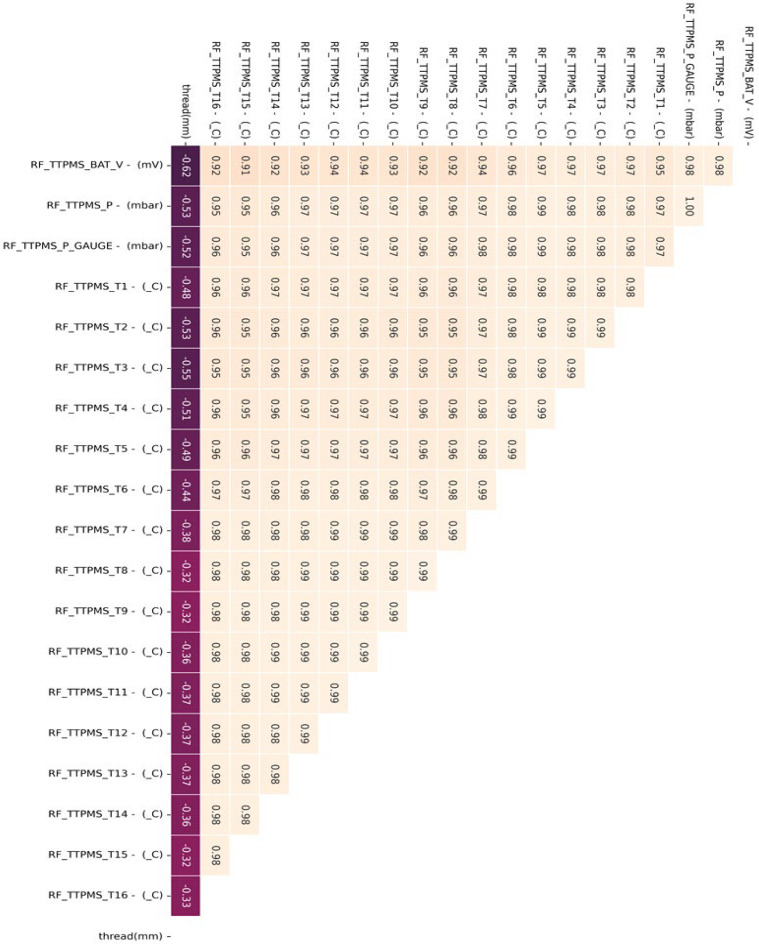
Correlation analysis of the metro tire measurement factors.

**Figure 7 sensors-23-08455-f007:**
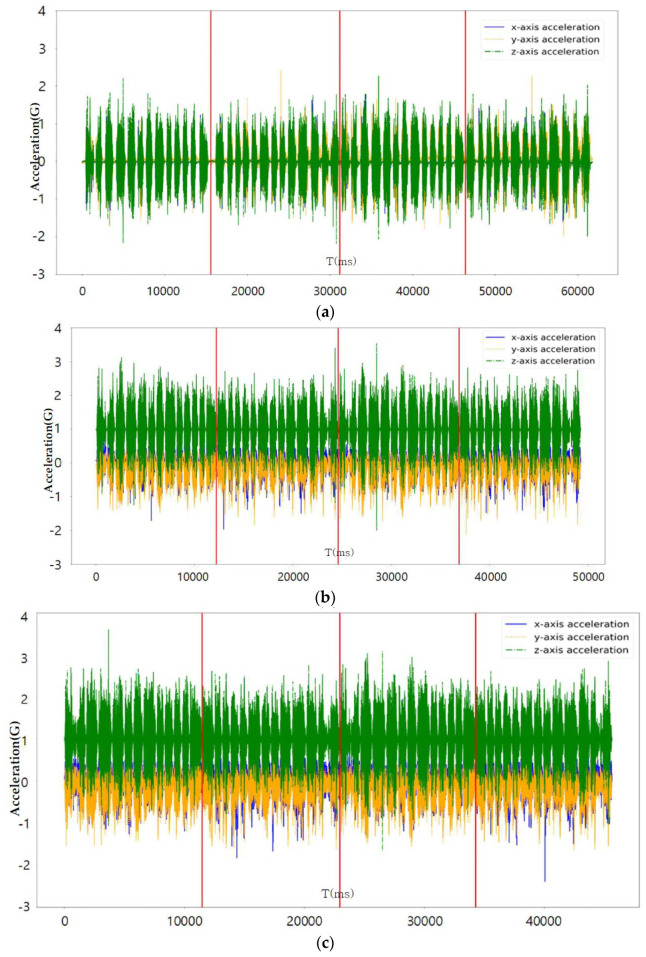
Axle acceleration measurement graph according to tire state: (**a**) Safety (tread 15 mm); (**b**) Warning (tread 8 mm); (**c**) Danger (tread 1.6 mm).

**Figure 8 sensors-23-08455-f008:**
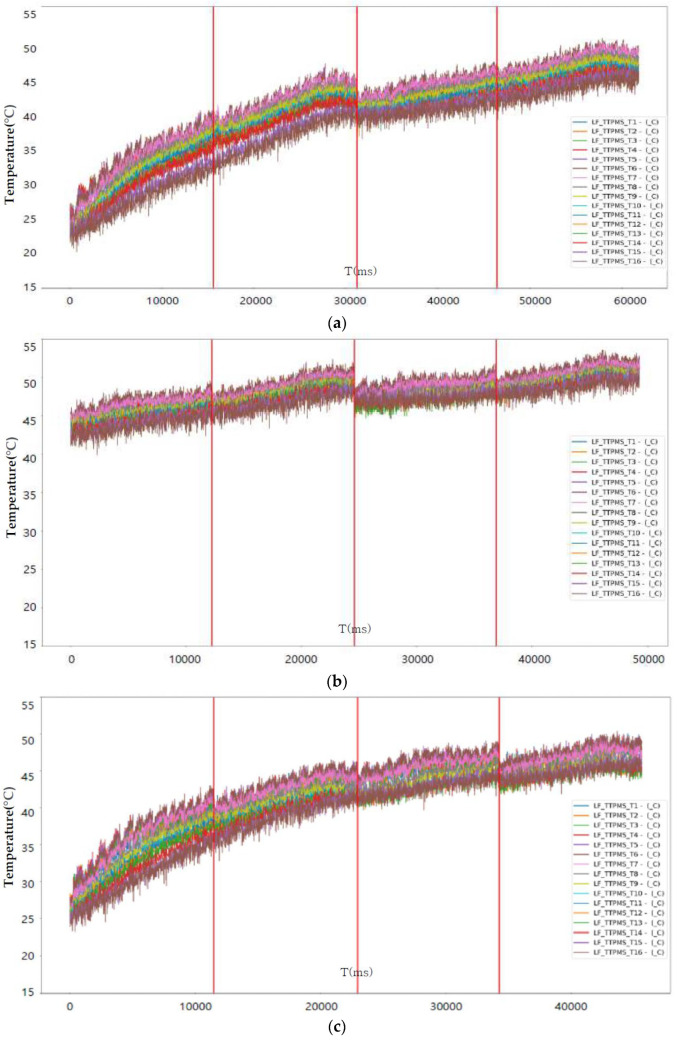
Internal temperature measurement graph according to tire (LF) state: (**a**) Safety (tread 15 mm); (**b**) Warning (tread 8 mm); (**c**) Danger (tread 1.6 mm).

**Figure 9 sensors-23-08455-f009:**
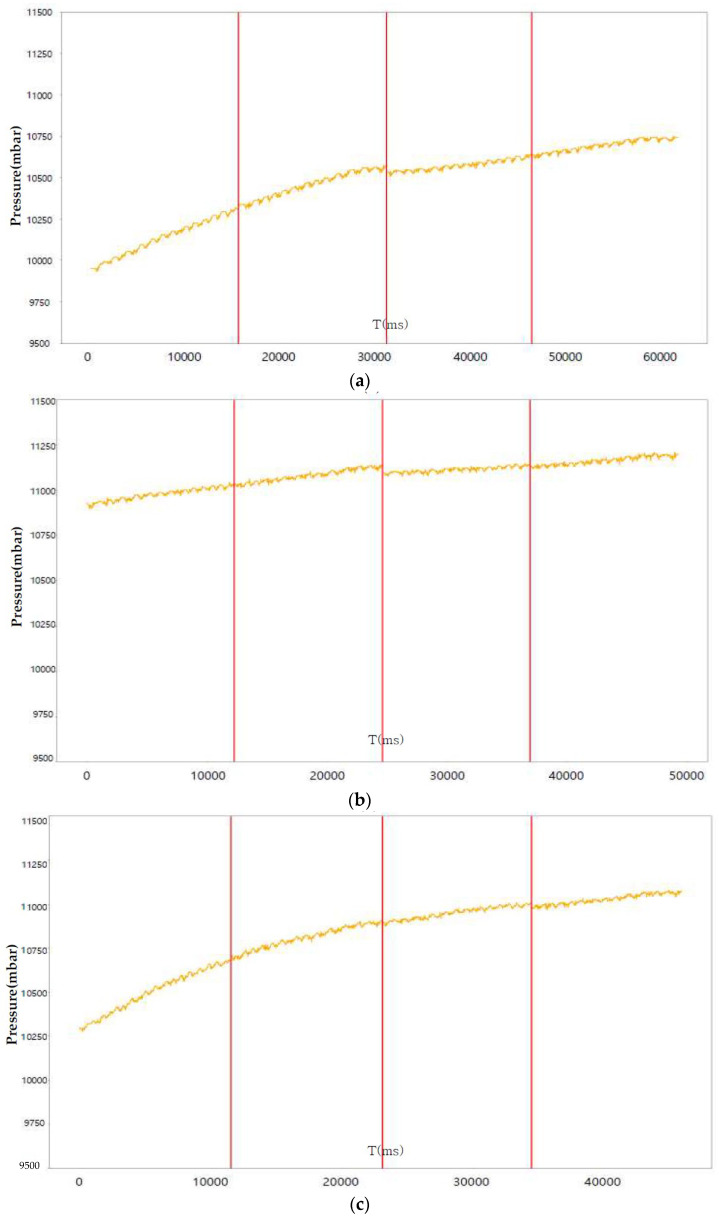
Internal pressure measurement graph according to tire (LF) state: (**a**) Safety (tread 15 mm); (**b**) Warning (tread 8 mm); (**c**) Danger (tread 1.6 mm).

**Figure 10 sensors-23-08455-f010:**
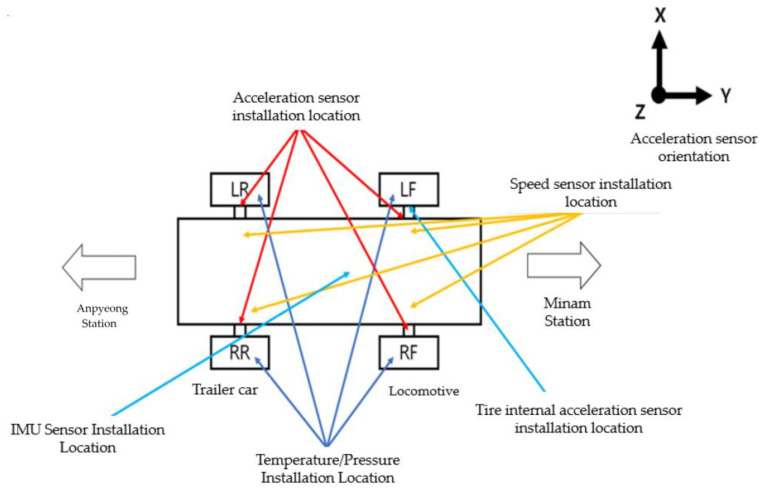
Location of installation by light rail measurement sensor on Busan Line 4.

**Figure 11 sensors-23-08455-f011:**
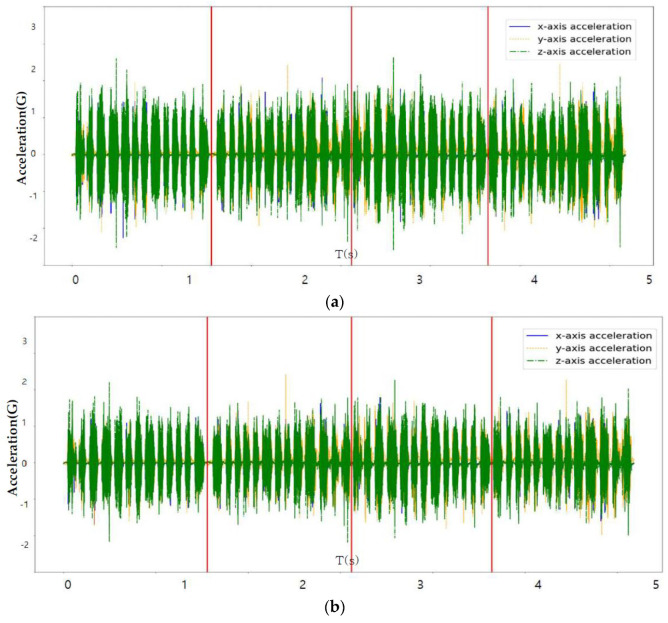
(Axle acceleration) Comparison of original learning data and sampling data: (**a**) Sampling: 1000 Hz; (**b**) Sampling: 100 Hz.

**Figure 12 sensors-23-08455-f012:**
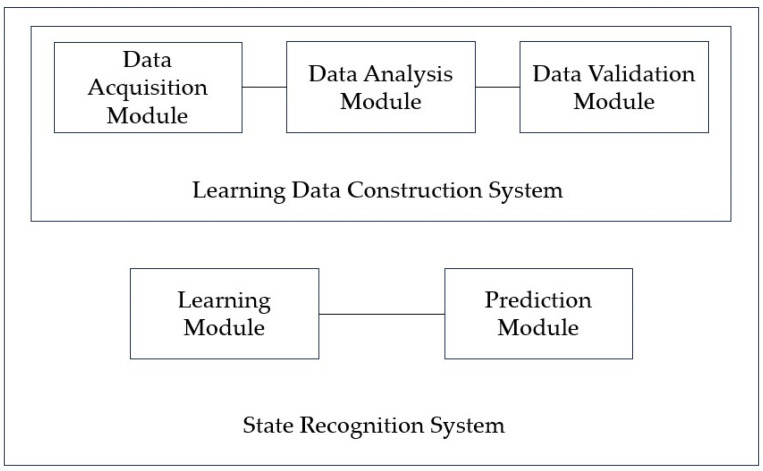
Block diagram of the device for state recognition of wheel member.

**Figure 13 sensors-23-08455-f013:**
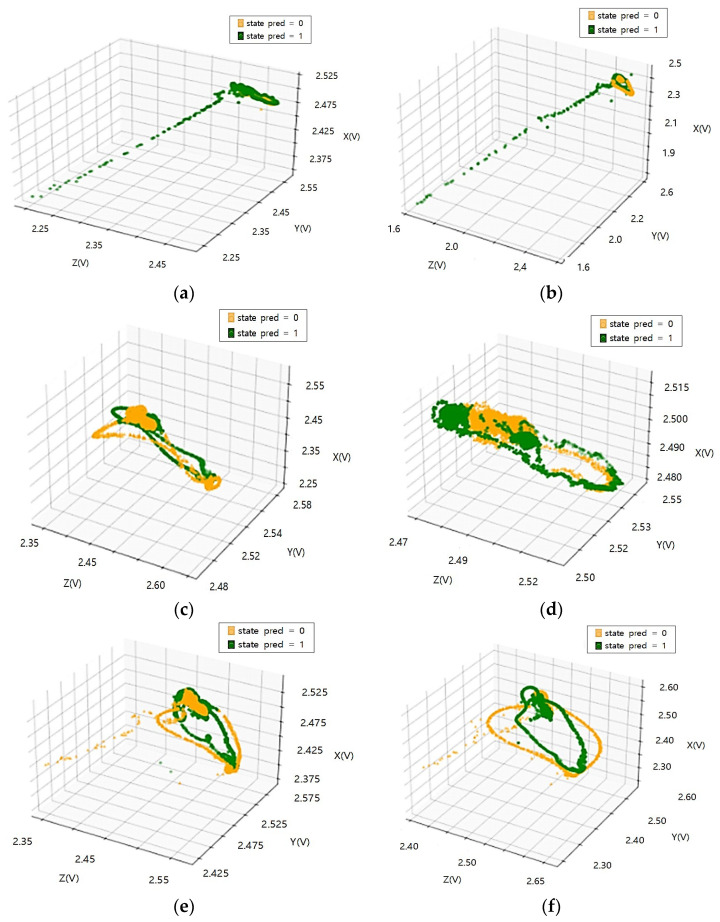
Tire state classification based on machine learning algorithms (SVM) using acceleration data: (**a**) [30,000 N_20 km] accuracy 77.74%; (**b**) [30,000 N_40 km] accuracy 79.91%; (**c**) [30,000 N_60 km] accuracy 78.61%; (**d**) [45,000 N_20 km] accuracy 63.64%; (**e**) [45,000 N_40 km] accuracy 70.44%; (**f**) [45,000 N_60 km] accuracy 71.14%.

**Figure 14 sensors-23-08455-f014:**
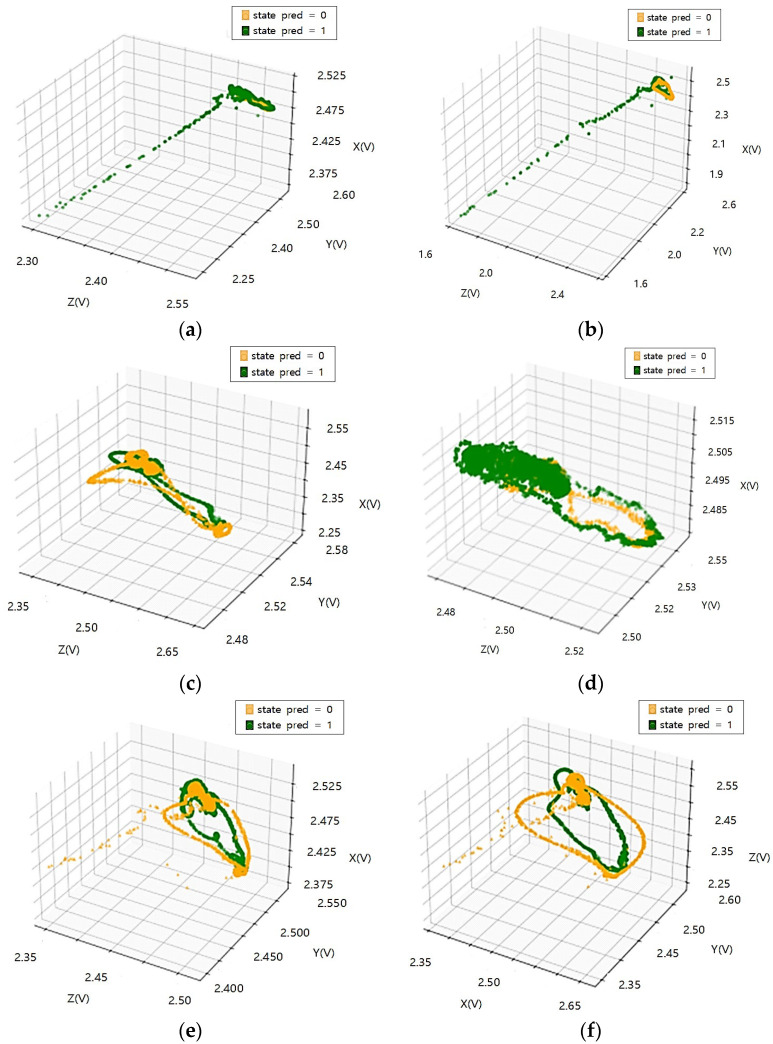
Tire state classification based on a machine learning algorithm (RF) using acceleration data: (**a**) [30,000 N_20 km] accuracy 88.12%; (**b**) [30,000 N_40 km] accuracy 91.37%; (**c**) [30,000 N_60 km] accuracy 94.53%; (**d**) [45,000 N_20 km] accuracy 91.37%; (**e**) [45,000 N_40 km] accuracy 88.49%; (**f**) [45,000 N_60 km] accuracy 93.38%.

**Table 1 sensors-23-08455-t001:** Types of factors affecting tire wear.

	Division	Main Factor	Contents	Note
1	Driving state	Slip angle	When vertical speed increases, the tread temperature increases.	kqq0=ΔHHΔqq0 *q*_0_: Original value of parameter. *q*: Variable quantity of parameters. *H*_0_: Original value of wear quantity. *H*: A variable quantity of wear quantity.
2		Vehicles speed	As vehicle speed increases, wear can also increase.	
3		Pressure	If the tire air pressure increases, it leads to energy loss.	
4	Non -driving state	Sprung mass	It is recommended to avoid overloading as a higher mass on the spring worsens tire wear.	
5		Ambient temperature	Ambient temperature affects tire wear.	

**Table 2 sensors-23-08455-t002:** Tire acceleration data collection environment information.

**☞ Test environment** · Data instrumentation: KUMHO TIRE · Test tire: 315/70 R20 RA04 Tire currently in use /worn tires in old state 2 copies · Test speed: 20/40/60 km/h · Test load: 3000/4500 kgf · Test air pressure: 130 psi · *X*-axis: driving direction, *Y*-axis: transverse direction, *Z*-axis: vertical direction · Sample rate: 1000 (Measurement for 1 min per test condition) · Data channel: 3 channels (acceleration 3 axes)	** 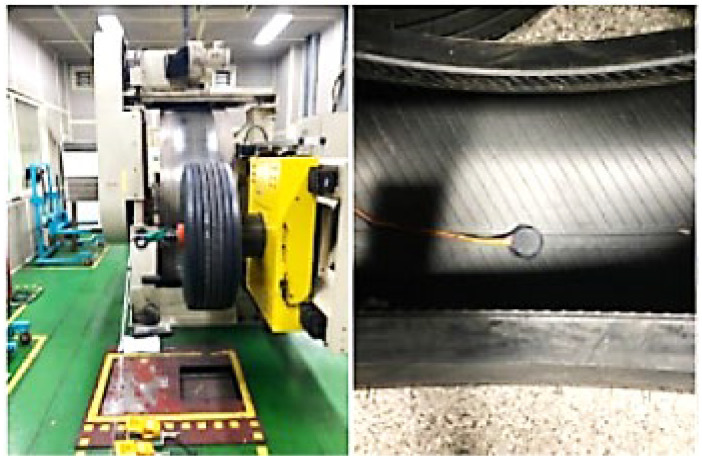 **

**Table 3 sensors-23-08455-t003:** Light rail transit driving measurement data information.

**☞ Test environment** · Data instrumentation : Busan Transportation Corporation · Test tire: safety (tread depth 14.5 mm) /Warning (tread depth 6–8 mm) /Danger (tread depth 1.6 mm) · Test speed: Operation speed within 60 km/h · Test air pressure: measured value · *X*-axis: driving direction, *Y*-axis: transverse direction, *Z*-axis: vertical direction (Composed of IMU and Axle-by-Axle acceleration Sensor) · Sample rate: 1000 (Measured during the operation time from the departure station to the destination station) · Data channel: 85 channels (August) → 153 channels (September)	** 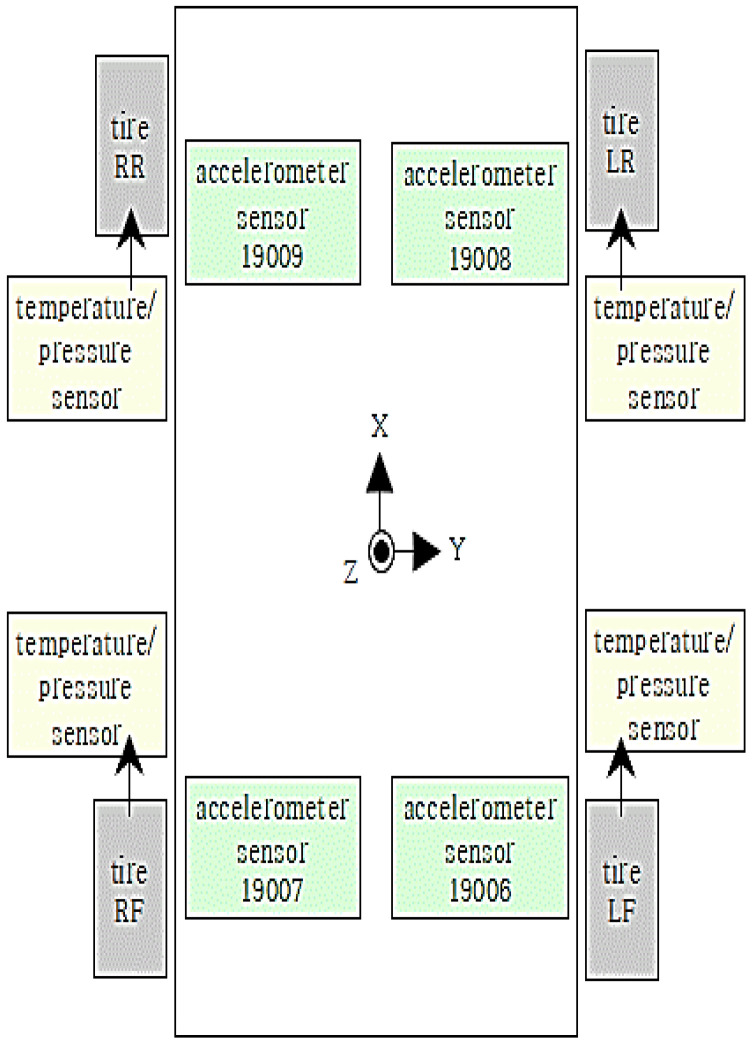 **

**Table 4 sensors-23-08455-t004:** Recognizing condition factor analysis applied channel (79 channels) and common acquisition state.

Data Division	Unit	Channel Name	Contents	Measurement in August	Measurement in September
				Tire State (Warning, Danger)	Tire Condition (Safety)
analysis factor	acceleration (g)	AI B-1~3	Axle acceleration of No. 3 (LF) (X, Y, Z)	O	O
	mV	LF_TTPMS _BAT_V	LF tire internal temperature/pressure sensor battery state (mV)	O	O
	pressure (mbar)	LF_TTPMS_P	LF tire pressure value	O	O
		LF_TTPMS_P _GAUGE	LF tire gauge pressure value	O	O
	Temperature (°C)	LF_TTPMS _T1~16	Temperature at 16 points inside the LF tire	O	O
	mV	RF_TTPMS _BAT_V	RF tire internal temperature/pressure sensor battery state (mV)	O	O
	pressure (mbar)	RF_TTPMS_P	RF tire pressure value	O	O
		RF_TTPMS_P _GAUGE	RF tire gauge pressure value	O	O
	Temperature (°C)	RF_TTPMS _T1~16	16 point temperature inside RF tire	O	O
	mV	LR_TTPMS _BAT_V	LR tire internal temperature/pressure sensor battery state	O	O
	pressure (mbar)	LR_TTPMS_P	LR tire pressure value	O	O
		LR_TTPMS_P _GAUGE	LR tire gauge pressure value	O	O
	Temperature (°C)	LR_TTPMS _T1~16	16 point temperature inside LR tires	O	O
	mV	RR_TTPMS _BAT_V	RR Tire Internal Temperature/Pressure Sensor Battery State (mV)	O	O
	pressure (mbar)	RR_TTPMS_P	RR tire pressure value	O	O
		RR_TTPMS_P _GAUGE	RR tire gauge pressure value	O	O
	Temperature (°C)	RR_TTPMS _T1~16	16 point temperature inside RR tires	O	O
tire state	mm	Tread	tire tread depth	O	O

**Table 5 sensors-23-08455-t005:** Configuration of learning data among light rail transit measurement data.

	Station Section	Measurement Date	Time	Number of Acquired Data	Tire State
1	Anpyeong → Minam	17 September	AM 06:50–AM 07:16	1,566,039	safety
2	Minam → Anpyeong	17 September	AM 07:18–AM 07:44	1,566,039	safety
3	Anpyeong → Minam	17 September	AM 07:51–AM 08:17	1,529,348	safety
4	Minam → Anpyeong	17 September	AM 08:20–AM 08:46	1,542,121	safety
5	Anpyeong → Minam	24 August	AM 06:42–AM 07:07	1,233,607	warning
6	Minam → Anpyeong	24 August	AM 07:11–AM 07:36	1,237,897	warning
7	Anpyeong → Minam	24 August	AM 07:44–AM 08:09	1,232,338	warning
8	Minam → Anpyeong	24 August	AM 08:13–AM 08:38	1,238,658	warning
9	Anpyeong → Minam	19 August	AM 07:34–AM 07:59	1,148,287	danger
10	Minam → Anpyeong	19 August	AM 08:03–AM 08:29	1,148,844	danger
11	Anpyeong → Minam	19 August	AM 08:34–AM 08:59	1,141,479	danger
12	Minam → Anpyeong	19 August	AM 09:05–AM 09:31	1,146,703	danger

**Table 6 sensors-23-08455-t006:** Tire recognizing condition algorithm learning data.

Tire State	(Anpyeong to Minam)	(Minam to Anpyeong)	(Anpyeong to Minam)	(Minam to Anpyeong)	Ratio
Safety	1,566,039	1,566,039	1,529,348	1,542,121	39.4%
Warning	1,233,607	1,237,897	1,232,338	1,238,658	31.4%
Danger	1,148,287	1,148,844	1,141,479	1,146,703	29.2%

**Table 7 sensors-23-08455-t007:** Learning data with sampling.

Tire State	(Anpyeong to Minam)	(Minam to Anpyeong)	(Anpyeong to Minam)	(Minam to Anpyeong)	Total Number of Data
Safety	156,604	156,604	152,935	154,212	620,355
Warning	123,361	123,790	123,234	123,865	494,250
Danger	114,829	114,884	114,148	114,671	458,532

**Table 8 sensors-23-08455-t008:** Removal of speed outlier data.

Tire State	(Anpyeong to Minam)	Minam to Anpyeong	Anpyeong to Minam	Minam to Anpyeong	Total Number of Data
Safety	977	359	966	234	2536
Warning	898	12	285	148	1343
Danger	0	88	988	84	1160

**Table 9 sensors-23-08455-t009:** Removal of tire internal temperature outlier data.

Tire State	(Anpyeong to Minam)	Minam to Anpyeong	Anpyeong to Minam	Minam to Anpyeong	Total Number of Data
Safety	2	0	1	0	3
Warning	50	52	48	50	189
Danger	47	46	54	877	200

**Table 10 sensors-23-08455-t010:** SVM (Linear Kernel) based Tire State Recognition result (Confusion Matrix).

			Predicted Tire State		
			Safety	Warning	Danger
actual tire state		safety	300,968	0	0
		warning	0	242,356	0
		danger	0	9506	179,639

**Table 11 sensors-23-08455-t011:** Tire State Recognition Result (Confusion Matrix) based on SVM (RBF Kernel).

			Predicted Tire State		
			Safety	Warning	Danger
actual tire state		safety	300,968	0	0
		warning	0	242,356	0
		danger	0	25,285	163,860

**Table 12 sensors-23-08455-t012:** Random Forest-based Tire State Recognition Result (Confusion Matrix).

			Predicted Tire State		
			Safety	Warning	Danger
actual tire state		safety	253,675	47,293	0
		warning	0	242,356	0
		danger	0	28,319	160,826

**Table 13 sensors-23-08455-t013:** The performance comparison result of machine learning-based tire recognizing condition algorithm.

Model	Performance Evaluation Index			
	Accuracy	Recall	Precision	F1 Score
SVM(Linear Kernel)	98.7%	98.7%	98.8%	98.7%
SVM (RBF Kernel)	96.5%	96.5%	96.9%	96.7%
Random Forest	89.7%	89.7%	92.1%	90.9%

## Data Availability

Not applicable.

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
