# Peer review of "A Study on Wheel Member Condition Recognition Using Machine Learning (Support Vector Machine)"

_sensors, 2023, doi:10.3390/s23208455_

Round 1
Reviewer 1 Report
This paper presents a method for predicting the abnormalities of railway wheels in advance and a method for improving the learning and prediction performance of the machine learning algorithm. It not only provided a method for predicting anomalies in railway vehicle wheels in advance, rather than relying on post-checks after anomalies occur but also demonstrated the potential for future applications and scalability in the expanding field of metro railways. Overall, the paper is well written and organized with a proper length. The contributions as well as the quality are both good.
1. The innovation of this paper is not clear and it is difficult for readers to understand the main contributions of this paper. This part should be added in Introduction section.
2. The description of the existing work should be shorter in Introduction section. Furthermore, more descriptions of the proposed method are needed.
3. In the main body of this paper, the collected data was analyzed to identify key factors that could be used for tire condition recognition (Z-axis acceleration data had a significant influence). Then in the following, "this study enhanced the learning and prediction performance of machine learning algorithms using temperature, pressure....". How to describe the relationship between the key factor and other factor in your machine learning based state monitoring design? Please give some explanations.
4. The reviewers recommend that more future work should be added on Conclusion Section. For example, some machine learning based data reduction algorithm, such as the Deep-PCA mentioned in the following reference in the filed of railway vehicles, can effectively enhancs the accuracy and performance of SVM. The authors should supplement some results on this aspect:
[1] Deep PCA-Based Incipient Fault Diagnosis and Diagnosability Analysis of High-Speed Railway
Traction System via FNR Enhancement. Machines, 2023, 11(4): 475.
Minor editing of English language required
Reviewer 2 Report
(1) This paper researches the wheel member condition recognition using SVM, but, from the structure and content of the whole paper, I don't see how SVM can identify wheel members. What is the specific SVM theoretical model method? What are the recognition methods and ideas of SVM? None of that is in the paper.
(2) The topic of this paper is to use SVM for recognition, and this paper mainly explains the machine learning method. How does the machine learning method relate to SVM?
(3) This paper lacks the description of necessary methods and recognition algorithms, but simply explains the results of recognition, which makes the credibility of the whole paper insufficient.
(4) Although the length of this paper is very long, there is still a certain gap with the writing of a real scientific paper. In addition, the figures in the paper are not clear.
Some English expressions need improvement
Reviewer 3 Report
Paper Summary: This paper presents a method for predicting the abnormalities of railway wheels in advance and a method for improving the learning and prediction performance of the machine learning algorithm. This paper showed that it can be applied to recognizing condition algorithms through characteristic analysis and correlation analysis of data distribution of acceleration factor. As a result of the experiment, it showed the highest recognition rate at 98.70% of SVM (Linear Kernel). I read the article, and this paper is well-written and contributes to the body of literature. However, I do have some major concerns with the current version that need correction during the revision. I believe the following comments can further enhance this paper's quality.
General concept comments:
1- The abstract doesn’t provide technical information on the proposed method. For example, what is data modality, what are the hyper-parameters, and why this method is imperative? Authors are advised to include such things in the revised work.
2- In the introduction, please write the implications (theoretical and technical significance) of your work in the contribution section by highlighting existing problems well. Also please write the contribution with bullets. In the current form, much of the contents are not succinctly written and do not concisely present the novelty of this paper.
3- It would be better to remove the figure from the introduction and place it in the proposed method section.
4- Figure 2, parts a and b, some portion in the x-axis is missing. Also, please convert the label of the y-axis into vertical form.
5- What are the axis labels in Figure 7? The same applies to Figure 8 and others.
6- The results are not compared with the existing methods. This is much needed task in any research paper.
7- Some more discussion about the result before the conclusion is desirable. Authors can provide some details of experiments and challenges that can stem from working on this kind of problem. Also, please compare your results with the existing work.
8- What is the uniqueness and novelty of this paper compared to existing work? In my opinion, many such works (e.g., fault diagnosis of machines or predictive maintenance) have already been proposed with extensive evaluations, and therefore, there is very little to no novelty in this work.
9- Limitations of this work are not given in the paper.
10- My final suggestion is to look at some good papers that have solved the same paper and organize the contents accordingly.
https://ieeexplore.ieee.org/abstract/document/9758642
https://ieeexplore.ieee.org/stamp/stamp.jsp?arnumber=8006280
https://ieeexplore.ieee.org/stamp/stamp.jsp?arnumber=9381866
English can be improved in some parts.
Round 2
Reviewer 2 Report
In this revised manuscript, the author has made a lot of improvements and improvements, the current manuscript is relatively clear, the author also added some explanations, the current manuscript can be published.
However, the author needs to make some modifications:
(1) The figure provided by the paper is still unclear, such as Figure 2-6 and Figure 13-14.
(2) The font in Figure 12 is not consistent with the font in the body. In short, the Figures of this paper needs to be re-presented with a more clear drawing.
(3)The algorithm in Figure 12 needs to give the processing steps.
The expression of the English language needs much improvement before publication
Reviewer 3 Report
The authors have addressed most comments. In the last round, please improve the figure's quality, formalization, and title of this paper. Also, please improve the abstract in terms of technical description.
Thanks
In some parts, English is a bit difficult to understand.
